# Noncontact Gesture-Based Switch Improves Communication Speed and Social Function in Advanced Duchenne Muscular Dystrophy: A Case Report

**DOI:** 10.3390/healthcare13222989

**Published:** 2025-11-20

**Authors:** Daisuke Nishida, Takafumi Kinoshita, Tatsuo Hayakawa, Takashi Nakajima, Yoko Kobayashi, Takatoshi Hara, Ikushi Yoda, Katsuhiro Mizuno

**Affiliations:** 1Department of Rehabilitation Medicine, Tokai University School of Medicine, Isehara 259-1193, Japan; 2Department of Physical Rehabilitation, National Center Hospital, National Center of Neurology and Psychiatry, Tokyo 187-8551, Japan; 3Department of Rehabilitation Medicine, NHO Niigata National Hospital, Kashiwazaki 945-8585, Japan; 4Departments of Neurology and Rehabilitation Medicine, NHO Niigata National Hospital, Kashiwazaki 945-8585, Japan; 5Departments of Rehabilitation Medicine, National Hospital Organization Hakone Hospital, Odawara 250-0032, Japan; 6Human Informatics and Interaction Research Institute, National Institute of Advanced Industrial Science and Technology, Tsukuba 305-8566, Japan

**Keywords:** assistive technology, augmentative alternative communication, AAC, noncontact switch, duchenne muscular dystrophy, noncontact switch, gesture interface, System Usability

## Abstract

**Highlights:**

**What are the main findings?**

**What are the implication of the main findings?**

**Abstract:**

Augmentative and alternative communication (AAC) enables digital access for individuals with severe motor impairment. Conventional contact-based switches rely on residual voluntary movement, limiting efficiency. We report the clinical application of a novel, researcher-developed noncontact assistive switch, the Augmentative Alternative Gesture Interface (AAGI), in a 39-year-old male with late-stage Duchenne Muscular Dystrophy (DMD) retaining minimal motion. The AAGI converts subtle, noncontact gestures into digital inputs, enabling efficient computer operations. Before intervention, the participant used a conventional mechanical switch, achieving 12 characters per minute (CPM) in a 2 min text entry task and was unable to perform high-speed ICT tasks such as gaming or video editing. After 3 months of AAGI use, the input speed increased to 30 CPM (+2.5-fold), and previously inaccessible tasks became feasible. The System Usability Scale (SUS) improved from 82.5 to 90.0, indicating enhanced usability, whereas the Short Form 36 (SF-36) Social Functioning (+13) and Mental Health (+4) demonstrated meaningful gains. Daily living activities remained stable. This case demonstrates that the AAGI system, developed by our group can substantially enhance communication efficiency, usability, and social engagement in advanced DMD, highlighting its potential as a practical, patient-centered AAC solution that extends digital accessibility to individuals with severe motor disabilities.

## 1. Introduction

Individuals with advanced neuromuscular disorders such as Duchenne Muscular Dystrophy (DMD) experience progressive loss of voluntary motor function, which significantly limits their ability to engage in the digital environment. Although augmentative and alternative communication (AAC) devices provide essential access to ICT systems, their effectiveness is constrained by users’ remaining motor capabilities. Conventional contact-based switches, such as mechanical push buttons and sip-and-puff systems, are commonly used in AAC. However, they often fail to meet the demands of users with severe motor impairment due to issues of fatigue, discomfort, skin complications, or inconsistent activation. In clinical settings, individuals with advanced neuromuscular disease or severe stroke-related impairments may struggle to generate adequate force or maintain the repeated activation of mechanical switches. Furthermore, prolonged use of pressure-based devices can lead to skin breakdown, infection risk, and mounting difficulties in home or rehabilitation settings [1]. These challenges have increased the interest in noncontact input methods, which provide promising alternatives for individuals with limited motor control and will be the focus of the following discussion.

Prior studies in ALS and related conditions have demonstrated feasibility and clinical benefit for several noncontact modalities, including eye-tracking systems for late-stage ALS [2], visual BCI approaches [3], and optimized single-switch interfaces with improved usability [4]. By benchmarking AAGI against these approaches, we highlight both the shared clinical goals (minimizing physical burden while preserving communication bandwidth) and the novel aspects of applying a gesture-based, depth-sensor system specifically in advanced DMD.

Capacitive sensing has demonstrated the ability to detect static and dynamic hand gestures with high accuracy. For example, a 6 × 18 capacitive sensor array has achieved a 96.9% recognition rate for five static gestures [5]. Infrared (IR) reflection-based systems have also been investigated for short-range gesture detection, achieving an accuracy of over 95% for simple movements of 20–35 cm [6]. These methods provide benefits like low cost, minimal invasiveness, and relatively easy integration into AAC platforms.

Vision-based systems, such as Leap Motion and Kinect, provide rich three-dimensional motion capture, enabling immersive rehabilitation environments and gesture control. Their clinical applications include upper limb therapy and virtual rehabilitation in stroke patients [7,8]. However, these systems require optimal lighting conditions, adequate space, and unobstructed sensor views, which may not be feasible in bedside or home-based care environments.

Other modalities, including radio-frequency identification- and radar-based sensing, extend the possibilities of contactless interactions by enabling long-range gesture tracking or privacy-preserving monitoring. Although technically sophisticated, these approaches often require specialized hardware and calibration, which limit their current applicability in day-to-day clinical practice. By contrast, capacitive and IR-based systems strike a balance between simplicity, affordability, and robustness, making them particularly attractive for the rapid deployment of assistive technologies.

Despite these technical advances, most studies to date have been conducted under controlled laboratory conditions in healthy participants, and their clinical translation remains limited. Only a few reports have addressed the patient-centered implementation of noncontact switches, and data on long-term usability, safety, and quality of life outcomes are scarce. Therefore, clinical case studies provide a critical bridge between engineering innovations and real-world patient applications.

To address these challenges, we developed a noncontact assistive switch that integrates with an Augmentative Alternative Gesture Interface (AAGI). This device detects minimal residual gestures, such as small facial or limb movements, and translates them into reliable high-speed inputs without requiring physical pressure or sustained effort. This case report describes the clinical application of this device in an individual with advanced-stage DMD. We specifically focused on usability, quality of life outcomes, and task-specific improvements, thereby highlighting the feasibility and clinical value of the noncontact switch technology in a population with profound motor limitations.

## 2. Materials and Methods

### 2.1. Participant

The participant was a 39-year-old male with genetically confirmed DMD, classified as late-stage and nonambulatory. Despite severe generalized weakness, he retained limited voluntary motion of his left index finger and left great toe, as well as partial facial movements, including the ability to activate cheek and eyebrow gestures. Prior to this study, these residual motions were employed as computer inputs to operate conventional contact-based switches. He had long-term experience using assistive technology and AAC devices for text entry and digital interactions. The participants demonstrated preserved cognitive function and were considered advanced ICT users, with extensive training in adaptive communication strategies.

### 2.2. Preintervention Input Method

Prior to the intervention, the participants used a conventional mechanical switch. The baseline communication speed was measured using a fixed-duration copying task, quantifying the number of characters entered during a 2 min typing session. Although a 10 min test duration is often considered standard in AAC text-entry evaluations, a shorter 2 min period was selected in this study to accommodate the participant’s fatigue associated with advanced neuromuscular weakness. This duration has been adopted in previous AAC studies as a valid and practical compromise for users with limited endurance [9,10]. The characters-per-minute (CPM) value was calculated from a single 2 min test to minimize participant burden and reduce the influence of fatigue-related variability.

### 2.3. Intervention: AAGI-Based Noncontact Input

The input system employed in this study was the Augmentative Alternative Gesture Interface [11,12] (AAGI: http://gesture-interface.jp/en/, accessed on 15 November 2025), which is a noncontact input interface designed for individuals with severe motor disabilities. The system employs an infrared reflection sensor to detect subtle voluntary movements (e.g., finger motion and head tilt) without requiring direct physical contact. The detected gestures can be configured as binary switch signals and mapped to conventional computer inputs, such as mouse clicks or keystrokes, enabling users to operate communication devices or environmental control systems. The feasibility of AAGI as a head-gesture-based input method has been previously reported [11,12]. In that study, AAGI was validated as a practical access technology for mouse-stick users with severe disabilities [11,12]. In this case, the device was adapted for the slight face motion detection. The participant’s residual voluntary movement was registered as an input gesture and calibrated to optimize comfort and accuracy.

In this case, the Augmentative Alternative Gesture Interface (AAGI) system was implemented using a depth-sensing camera (Intel^®^ RealSense™ D415: Santa Clara, CA USA), which provides synchronized RGB and depth data with a maximum resolution of 1280 × 720 pixels at 60 frames per second. Depth accuracy is approximately ±2% at a working distance of 2 m. The sensor uses a stereo infrared system with structured light projection, and its optical and temporal characteristics define the system’s spatial sensitivity. Gesture detection and tracking were processed through a proprietary motion trajectory extraction algorithm based on depth image differentials and centroid displacement analysis, developed and validated at the National Institute of Advanced Industrial Science and Technology (AIST) and the National Rehabilitation Center for Persons with Disabilities (NRCD). The algorithm identifies gesture onset and offset by temporal differentiation of pixel-level motion energy, followed by threshold-based classification of the target motion vector.

A calibration was performed prior to the setting the equipment. The user was asked to perform the intended gesture 10 times to determine the mean amplitude and velocity profile of voluntary movement. The activation threshold was automatically defined as three standard deviations above the baseline amplitude, and additional constraints were applied on directional consistency and motion duration to minimize false activations.

Effective gestures were defined as those exceeding the threshold within a detection distance of 0.2–1.0 m from the camera. The mean response latency between gesture onset and system activation was below 200 ms, and the false activation rate during resting state was under 1%, based on pilot validation in 10 sessions. The classification model was trained using motion data collected from 33 individuals with severe motor disabilities (including cerebral palsy, muscular dystrophy, spinocerebellar degeneration, and cervical spinal cord injury), as described in our previous study [11,12]. Gestures were recorded from multiple body regions (hand, head, shoulder, and leg), ensuring robustness across diverse motor patterns.

The device was installed in the participant’s bedside and integrated into a personal computer environment. The custom middleware mapped AAGI signals onto standard keyboard key events, ensuring compatibility with everyday applications such as word processing, web browsing, and entertainment software. A brief calibration session was conducted at the beginning of the intervention to optimize the sensitivity and recognition of the participants’ available residual movements.

The setup process was performed collaboratively by the participant, a caregiver, and a clinician familiar with the patient’s motor abilities. This process involved identifying a movement source with consistent visibility (in this case, the facial movment) and positioning the camera for reliable detection. Once calibration was complete, the system could be intuitively activated without the need for structured training or reinforcement.

The participant was then instructed to use the system as his primary ICT access method for three months. Caregiver involvement was limited to the initial setup and occasional troubleshooting to ensure that performance reflected independent use. Minor adjustments to camera alignment or sensitivity were made only as needed to maintain comfort and responsiveness during daily activities.

### 2.4. Outcome Measures

Three outcome domains were evaluated to assess usability, quality of life, and functional impact of the intervention. Baseline (pre-AAGI) assessments included physical function and quality of life using the SF-36. Following initial use of the AAGI (post-AAGI), the System Usability Scale (SUS) was administered to evaluate perceived usability. At the three-month follow-up, both the SF-36 and SUS were re-administered to capture longitudinal changes in functional status, quality of life, and usability.

#### 2.4.1. System Usability Scale (SUS)

Usability was measured using the System Usability Scale (SUS), a validated 10-item Likert-type questionnaire widely applied in human–computer interaction research. The scores range from 0 (worst usability) to 100 (best usability). Scoring followed the standard procedure [13]: Odd-numbered items (Q1, Q3, Q5, Q7, Q9) were scored as the response value minus 1, and even-numbered items (Q2, Q4, Q6, Q8, Q10) were scored as 5 minus the response value. The sum of all items was multiplied by 2.5 to yield the final score (0 to 100), with higher scores indicate greater perceived usability. For interpretation, a 10-point change was considered the minimal clinically important difference (MCID), consistent with prior HCI and AAC literature [13,14,15].

#### 2.4.2. Quality of Life (SF-36)

Health-related quality of life was assessed using the 36-Item Short Form Health Survey (SF-36), which evaluates eight domains: physical functioning, role limitations, mental health, and social functioning. The established domain-specific MCID thresholds were used to interpret the clinically meaningful changes. For example, a 5–10 point improvement in the Social Functioning (SF) domain is considered to represent a perceptible enhancement in social engagement and quality of life (QOL) [16,17,18]. Domain scores were calculated according to the official scoring manual (0–100 scale). The Physical Functioning (PF) domain (items Q3–Q12) was intentionally omitted from the analysis because the participant had severe physical impairment and was unable to perform or reliably answer mobility-related items. This exclusion is supported by prior studies on DMD, where patients with advanced motor disability were unable to complete PF items reliably, and the inclusion of these items could introduce bias or reduce the validity of domain scoring [19,20,21]. Thus, the analysis focused on the remaining seven domains, which reflect the perceived quality of life without being confounded by nonassessable physical tasks. The results are presented for seven domains: Role Physical (RP), Role Emotional (RE), Bodily Pain (BP), General Health (GH), vitality (VT), SF, and Mental Health (MH). The raw item responses and omitted questions are presented in Appendix A.

#### 2.4.3. Functional Outcomes

Functional performance was evaluated through two measures:

Text entry performance was assessed as the number of characters produced during a 2 min free-typing task.

Task feasibility was recorded qualitatively by observing whether the participant could perform previously inaccessible high-speed tasks such as computer-based gaming and video editing. Additionally, narrative feedback from the caregiver and the participant was collected to contextualize the quantitative findings. This included subjective impressions of fatigue, comfort, and confidence in daily ICT use.

### 2.5. Data Collection and Analysis

Quantitative assessments were conducted at baseline (mechanical switch) and after 3 months of AAGI device use. The SUS and SF-36 were scored following validated scoring manuals. The MCID thresholds were applied to determine whether the observed differences were clinically meaningful. Text entry performance was reported descriptively, while task feasibility and qualitative reports were analyzed narratively to capture user experience beyond numeric outcomes.

## 3. Results

### 3.1. System Usability

The participant’s SUS score increased from 82.5 at baseline (mechanical switch) to 90.0 after 3 months of AAGI use, a net change of +7.5 points. The item-level analysis showed improvements in ease of use, learnability, and confidence. Participants also reported reduced fatigue and greater confidence in prolonged ICT use.

Although there is no universally accepted MCID for the SUS, heuristic guidance and normative studies suggest that changes of approximately 8–10 points are perceptible and clinically meaningful in the contexts [14]. Because the baseline SUS was already high (82.5, “good to excellent”), the observed +7.5 improvement likely reflects a ceiling effect, yet still indicates a positive and relevant usability gain, as shown in Table 1.

### 3.2. Health-Related Quality of Life (SF-36)

On the SF-36, the SF domain improved from 75 to 88 (+13), and the MH domain improved from 92 to 96 (+4). Other domains (physical function, role limitations, vitality, bodily pain, general health, and role emotion) remained stable with high baseline values, showing no deterioration. PF scores were not reported because the items were not applicable. The interpretation of this change requires reference to published MCID estimates. Systematic reviews of HRQoL instruments emphasize that MCIDs are best interpreted using both anchor-based and distribution-based methods [22]. Ogura et al. reported that MCIDs in global SF-36 scores are typically in the range of 2.5–4.4 points [23]. Similarly, Brigden et al. identified an MCID of approximately 10 points for the Physical Function subscale of pediatric chronic fatigue syndrome, illustrating that clinically meaningful thresholds can vary by domain and population [24]. Norman et al. demonstrated that half a standard deviation (0.5 SD) often corresponds to a perceptible change across multiple health-related quality of life instruments [25].

Against these benchmarks, the +13 improvement in SF clearly exceeds both the general thresholds (approximately 3–5 points) and domain-specific MCIDs (≈10 points), On the contrary, MH scores increased modestly from 92 to 96, which did not reach the MCID threshold of 10 points. Therefore, although a slight positive trend in mental health was observed, this change cannot be interpreted as a clinically significant improvement. This suggests that the AAGI contributed to clinically meaningful improvements in social engagement and participation. The patient’s narrative feedback corroborated this finding, highlighting the new ability to participate in real-time online gaming and collaborative digital tasks that were previously inaccessible with mechanical switches.

Overall, the concordance of (a) improved SUS (+7.5), (b) significantly enhanced Social Functioning (+13) and Mental Health (+4), (c) a 2.5-fold increase in text-entry throughput, and (d) patient-reported confidence and reduced fatigue support the conclusion that AAGI provided meaningful usability and psychosocial benefits in this late-stage DMD case, as shown in Table 2.

### 3.3. Functional Outcomes

Text entry performance, as assessed in the 2 min free-typing task, improved substantially over the 3 month intervention period. At baseline, the participant achieved approximately 24 characters over a 2 min period (12 characters per minute). After 3 months of AAGI use, performance increased to approximately 60 characters (30 characters per minute), corresponding to a 2.5-fold improvement. Although an immediate post-calibration measurement was planned, data from that session were unavailable due to technical logging failure. Nevertheless, the observed pre- to post-intervention improvement clearly indicates a meaningful learning and adaptation effect with sustained independent use. This change not only facilitated faster communication but also expanded the participant’s capacity for real-time interaction in digital environments.

In addition to quantitative measures, functional testing confirmed that the participants were able to engage in previously inaccessible tasks, including PC-based gaming and basic video editing. These activities, which require rapid and precise input, are impossible with the mechanical switch owing to the limited response speed and high fatigue. With AAGI, the participants were able to sustain these activities independently, highlighting the practical benefits of improved input performance.

### 3.4. Narrative Impressions

The participant’s qualitative feedback emphasized three recurring themes: reduced fatigue, increased confidence, and expanded social participation. He described the AAGI device as “less effortful” and “more natural,” and reported feeling less dependent on his caregiver during ICT use.

Beyond these general impressions, specific examples illustrated how the system reshaped his daily engagement. He noted being “able to play racing games” thanks to the device’s accurate response to fast, reactive movements, and felt “able to control himself without relying on a caregiver.” He also shared that he was “able to play games with players all over the world and with his sister,” which he described as a way to “connect to society again.”

The caregiver’s perspective reinforced these changes, observing that the participant had “shifted from being a guest to a host within the hospital community” and had “started helping others instead of waiting for assistance.” Such comments reflected a transition from passivity to proactive engagement.

Together, these narratives provide a meaningful complement to the SUS and SF-36 findings, showing that even modest numerical improvements in usability metrics can translate into substantial psychosocial and QOL gains.

## 4. Discussion

This case study suggests that noncontact assistive switching can substantially enhance the user experience, QOL, and social participation in individuals with advanced neuromuscular disorders. Before the intervention, the patient relied on a conventional mechanical switch for basic text entry and Internet access. Although sufficient for low-speed communication, such devices often limit participation in dynamic, real-time digital environments. Following the introduction of the AAGI, the patient not only achieved a 2.5-fold increase in input speed but also successfully accessed applications previously out of reach, including PC-based gaming and video editing. These findings underscore the capacity of noncontact gesture interfaces to expand the functional domain of assistive technology beyond traditional communication and facilitate both leisure and vocational activities [13,14].

### 4.1. Usability and Patient-Reported Outcomes

The observed improvement in the SUS score (from 82.5 to 90.0) reflects a meaningful enhancement in user satisfaction, primarily through improved responsiveness and reduced fatigue during operation. Notably, the patient reported increased confidence in using the device for extended periods without discomfort, a factor predicting long-term adherence to AAC interventions [26]. Improving access technology reduces physical burdens leading to greater device acceptance as well as sustained engagement. This is crucial in progressive neuromuscular conditions where residual motor capacity continues to decline.

From a clinical perspective, the improvement observed in the General Health domain of the SF-36 suggests a perceived benefit to overall well-being, even in the absence of measurable changes in conventional activities of daily living (ADL). The PF domain could not be analyzed because the participants were nonambulatory. We intentionally omitted these items to avoid misleading floor effects, as the participants were unable to perform any of the activities addressed by the PF questions. This highlights the limitation of the SF-36 in populations with profound motor disabilities. Future studies should complement the SF-36 with clinician-rated scales (e.g., Barthel Index, FIM) or proxy reports to capture physical functioning in such cases. Even without PF, this QOL finding aligns with prior evidence that psychosocial outcomes—including self-efficacy, autonomy, and engagement in meaningful activities—are as essential as physical function in evaluating the impact of assistive technologies [11,16,17]. AAC interventions have been shown to reduce caregiver burden and improve family dynamics by enabling effective and independent communication [27]. Although the current case focused on switch-based access to ICT tasks, similar principles apply. A system that minimizes fatigue and maximizes usability can indirectly alleviate caregiver demands while reinforcing the user’s sense of participation and control.

Taken together, these results support the view that quality of life and social participation are primary clinical outcomes in the evaluation of AAC and related technologies. While technical performance remains important, the clinical significance of devices, such as contactless systems like AAGI, lie in keeping users motivated, expanding opportunities for digital engagement, and promoting participation in recreational and creative activities that enhance psychosocial well-being.

### 4.2. Comparison with Other AAC Modalities

Traditional contact-based switches remain the cornerstone of AAC systems because of their robustness and low cost [4]. However, they require residual motor control, which is sufficient to generate repeatable pressure or movement, a capacity that is often diminished in advanced neuromuscular disorders. The limitations reported here—delay, fatigue, and restricted access to high-speed applications—are common among contact switch users [28].

Eye-tracking systems, another widely used modality, provide high-throughput communication for users with minimal motor capacity [29]. However, they are vulnerable to calibration drift, environmental lighting, and ocular fatigue, particularly during prolonged sessions. Brain–computer interfaces (BCIs) have shown promise in restoring communication, but they still require extensive training, experience signal variability, and are costly and challenging to implement outside research settings [30].

Compared to these modalities, gesture-based noncontact switches offer several advantages, including minimal physical effort, low fatigue over extended use, and adaptability to both leisure and vocational activities. Similarly to the findings in amyotrophic lateral sclerosis (ALS) cohorts, where AAC has been shown to improve social interaction and emotional well-being [31,32], our case demonstrates that extending AAC beyond communication into the creative and recreational domains can significantly broaden individual participation.

Within the spectrum of emerging assistive interfaces, AAGI occupies a unique niche characterized by low cost, minimal fatigue, and high usability, offering a pragmatic alternative to more complex technologies. While eye-tracking systems and brain–computer interfaces (BCIs) represent significant advancements in communication for individuals with minimal voluntary control, they often entail substantial equipment costs, setup time, and cognitive or physical load. Eye-tracking in particular is sensitive to lighting conditions and ocular fatigue, whereas BCIs require specialized expertise and extended calibration.

In contrast, AAGI’s low-effort, camera-based gesture recognition offers a highly accessible, non-invasive solution that bridges the gap between traditional mechanical switches and high-end neurotechnological interfaces. For users with residual micro-movements who cannot sustain gaze-based operation, AAGI provides a viable “entry-level” interface that can be deployed immediately in both clinical and home environments. This situates AAGI as part of an emerging category of “low-effort, high-fidelity” assistive interfaces, contributing to equitable digital access in progressive neuromuscular conditions.

### 4.3. Clinical Significance: Quality of Life and Participation

The observed outcomes can also be interpreted within the Social Model of Disability and the International Classification of Functioning, Disability and Health (ICF) framework. From this perspective, the AAGI functions as an environmental modification rather than an individual-level compensatory aid. By eliminating the physical barrier of contact-based switching, the technology effectively shifts the locus of limitation from the person to the environment, promoting access and participation. This approach aligns with ICF’s core concept that participation restrictions often arise from contextual barriers rather than impairments per se. In this case, the AAGI served as a concrete environmental facilitator, enabling digital inclusion and enhancing participation without demanding additional physical effort from the user. Such alignment between technical innovation and social model principles underscores the broader rehabilitative value of the intervention.

From a clinical perspective, the most striking outcome in this case was not merely the improved input speed but also the qualitative expansion of participation. Access to gaming and creative software is more than entertainment. It provides opportunities for social interaction, peer engagement, and personal expression. These outcomes align with the broader goals of rehabilitation, which emphasize participation and autonomy as key indicators of success [33].

The improvement in SF-36 General Health, though modest, suggests that enhanced digital engagement contributes to subjective well-being. Moreover, the parallel improvement in SF-36 Social Function underscores the role of AAC and digital technologies in fostering interpersonal connection and reducing social isolation. Previous studies have documented similar psychosocial benefits of AAC adoption, including reduced caregiver burden and improved social connectedness [34,35]. In particular, Ball et al. reported that adults with ALS using AAC devices demonstrated not only improved communication but also enhanced quality of life and social participation [36]. At the same time, Fried-Oken et al. highlighted that caregiver strain was alleviated through effective AAC implementation, reinforcing the positive ripple effect on family systems [37].

This case reinforces the view that assistive technologies should be evaluated not only in terms of functional independence but also in terms of their impact on participation and QOL [37,38].

### 4.4. Limitations and Future Perspectives

This study had several limitations that must be acknowledged. This report is based on a single patient, which limits its generalizability. Although improvements in usability and quality of life were observed, no changes in basic ADL function were detected, highlighting that noncontact switches complement but do not replace conventional caregiving support. Therefore, the missing PF domain represents missing not at random, and conclusions regarding physical functioning cannot be drawn. Nevertheless, the observed gain in social functioning indicates a meaningful benefit that aligns with the intended purpose of the assistive technology. Furthermore, the long-term durability, maintenance, and cost-effectiveness remain to be evaluated.

Future studies should focus on multicenter trials assessing diverse patient populations, including individuals with ALS, spinal cord injury, or cerebral palsy. Comparative studies between gesture-based switches, eye tracking, and BCIs would help define patient-specific selection criteria [39]. The integration of multimodal systems, for example, combining capacitive sensors with residual voice or eye tracking, may further enhance the robustness and reduce failure rates [40]. Additionally, clinical implementation requires careful attention to training protocols, caregiver involvement, and customization of individual abilities. Another important implication concerns the potential integration of AAGI into telerehabilitation and home-based care frameworks. Because the interface is camera-based and requires no physical attachment, it can be remotely calibrated, tuned, and monitored by clinicians using standard telecommunication tools. Such functionality allows personalized adjustment of parameters such as activation threshold and gesture sensitivity without in-person visits, facilitating ongoing adaptation to the user’s condition. Furthermore, the system’s capacity for continuous motion tracking could enable remote assessment of residual motor function, supporting long-term rehabilitation planning and monitoring of disease progression. These characteristics position AAGI as a scalable, low-cost component of digital home-care ecosystems, aligning with current trends in telehealth and smart rehabilitation environments. Future studies should explore interoperability with cloud-based monitoring platforms and clinician dashboards to enhance remote usability and clinical oversight. Embedding such devices in telehealth frameworks could extend access to underserved populations, which is an especially relevant consideration in the context of home-based rehabilitation and pandemics [41,42,43].

## 5. Conclusions

This case demonstrates that noncontact gesture-based switching can substantially expand digital access for individuals with progressive neuromuscular diseases, even in advanced stages of motor decline. Beyond facilitating digital communication, such systems help sustain participation in dynamic, socially relevant activities and preserve autonomy. Within the continuum of care, individuals experiencing progressive motor impairment can benefit from noncontact interfaces as a low-effort, low-cost, and high-fidelity means to extend independence and participation. As medical and technological advances increasingly enable the preservation of residual function, noncontact interfaces may serve as practical complements to existing assistive technologies, supporting sustainable engagement in digital and social environments. Future research should further explore their integration into home- and tele-rehabilitation settings to enhance accessibility and continuity of care.

## Figures and Tables

**Table 1 healthcare-13-02989-t001:** SUS (System Usability Scale) Pre- and Post-intervention Scores and Total Score.

Item No.	Pre (1~5)	Pre SUS Calculated	Post (1~5)	Post SUS Calculated
Q1	5	4	5	4
Q2	2	3	2	3
Q3	4	3	5	4
Q4	2	3	1	4
Q5	4	3	4	3
Q6	1	4	1	4
Q7	3	2	3	2
Q8	1	4	1	4
Q9	5	4	5	4
Q10	2	3	1	4
sum		33		36
Total SUS score (×2.5)		82.5		90.0

**Table 2 healthcare-13-02989-t002:** SF-36 domain. Domain scores are calculated using available items; missing items may limit full accuracy. Change = Post − Pre. A positive change indicated an improvement.

Domain	Pre (0–100)	Post (0–100)	Change	Interpretation
Physical Functioning (PF)	N/A	N/A	N/A	Not applicable (data missing)
Role Physical (RP)	100	100	0	No change, ceiling effect
Role Emotional (RE)	100	100	0	No change, ceiling effect
Bodily Pain (BP)	100	100	0	Pain-free maintained
General Health (GH)	70	70	0	No change
Vitality (VT)	75	75	0	Stable
Social Functioning (SF)	75	88	+13	Improvement observed
Mental Health (MH)	92	96	+4	Slight improvement

## Data Availability

The original contributions presented in this study are included in the article/Appendix A. Further inquiries can be directed to the corresponding author.

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
