# Peer review of "Noncontact Gesture-Based Switch Improves Communication Speed and Social Function in Advanced Duchenne Muscular Dystrophy: A Case Report"

_healthcare, 2025, doi:10.3390/healthcare13222989_

Round 1

Reviewer 1 Report

Comments and Suggestions for Authors

Reviewer Comments and Suggestions for a Minor Revision

The manuscript describes a compelling case report of a clinical intervention using a Noncontact Gesture-based Switch (AAGI) to enhance communication and social function in an individual with advanced Duchenne Muscular Dystrophy (DMD). The work is of interest to the rehabilitation medicine and assistive technology (AT) communities and describes an innovative and generally low-level posture of support for a population with extremely limited options for movement.

I believe the finding of a doubling of input over the 3-month period is a strong quantitative data collection. The suggested revisions below are intended to enhance the manuscript’s coherence and theoretical backing prior to publication. The authors have provided a unique, effective clinical intervention, and these revisions will help solidify the work’s place within the AT literature.

Specific Comments by Section

Introduction

To enhance the theoretical context, the authors can briefly relate the need for the AAGI to the Technology Acceptance Model (TAM). For advanced DMD patients, the Perceived Ease of Use (a key component of TAM) for current switches (i.e., einp-and-puff or conventional button switches) becomes very low due to muscle weakness. This framing represents a robust academic foundation.

The existing literature review is fine but should also include recent works addressing non-contact/minimal effort interfaces in other similar rapidly progressive neuromuscular diseases (for example, advanced ALS) for the purpose of benchmarking the novelty of such an application in DMD.

Case Presentation and Methods

The authors are to be commended for having selected an appropriate methodological framework (Case Report) to introduce this new intervention. Furthermore, the objective measurement of input speed is a methodological sound approach. To ensure maximum replicability (again for the AAGI and the input speed for a given condition) and scientific rigor, I suggest the following methodological refinements:

Technical and Calibration Detail: The authors have fully described the AAGI system and the patient in detail. For replicability, can you provide more detail on the technical specifications (e.g., minimum camera resolution, type of vision algorithm)? Along those lines, please describe how you established the movement threshold to differentiate minimal voluntary from involuntary spasms or tremor, which are common in these conditions?

Training Protocol: Please provide clarification on the training protocol – how often were they trained, what duration did training typically entail, and were any specific reinforcement methods used to enforce consistency in the activation gesture?

Baseline Measures: Please clarify the method used for baseline assessment. What specific standardized communication or Quality of Life (QoL) measures were used to establish the pre-AAGI status? If specific measures were utilized, naming them, if appropriate, would add rigor to the pre-post assessments and subscription outcomes.

Results

The presentation of the speed metrics for input is excellent. You might approach the QoL and social function results more forcefully by making better use of qualitative data. Including patient or caregiver quotes that were de-identified and specific helped with the anecdotal outcome of social activity change (e.g., "was able to complete a purchase online independently," "had the opportunity to participate in a group chat for the first time").

Also, ensure that when providing figures around speed change, it is easy to identify the baseline and final measurement. A simple line graph that depicts the trajectory of improvement over the three-month period or reminds readers that a learning curve has been created would show learning from pre to post.

Discussion

This is the area where the most robust academic significance can be generated. The authors should definitely consider linking the outcome in practice to the applicable theory.

Connection between Theory and Practice - Social Model: A substantial addition would be to relate the success of the AAGI to the Social Model of Disability. The technology served as a crucial environmental modification, removing a physical barrier (the requirement for switch contact) to digital and social participation. The intervention shifts the focus of the impairment (muscle weakness) away from the individual to an environmental barrier, thus enhancing Participation, a core outcome in the ICF classification system.

 Compare Emerging Trends - Niche: The discussion can further compare the AAGI's role to the considerably more complicated higher cost emerging trends/systems like Brain-Computer Interfaces (BUI) or highly advanced Eye-Tracking Systems. The AAGI is an extremely important non-invasive option in the ‘low-effort, low-cost, high-fidelity’ niche. It is a critical starting point for patients that have so little remaining movement to physically operate a switch, yet still express eye fatigue (thus are unable to perform eye-tracking movements) or do not have BUI options nor the ability to interface with one.

Implications Telerehabilitation - In Implications it should discuss the anticipation that this camera interface could be easy to integrate into Tele-Rehabilitation models. The set-up includes visual tracking and requires minimal physical interaction with the patient, which would allow for the remote calibration, tuning, and potentially monitoring of switch set-up parameters, a promising future application for the home-care segment.

Conclusions

The Conclusion should be re-evaluated for emphasis and provide a clear, prescriptive recommendation. Wording should be suggested to establish AAGI as an important consideration in a continuum of care for patients with progressive neuromuscular diseases when the patient reaches the point of progressive functional decline where other input options can no longer be used.

Author Response

We greatly appreciate your constructive suggestions, which have helped us to substantially improve the clarity, methodological rigor, and overall quality of the paper.

All comments were carefully considered and addressed in the revised manuscript. Revisions are highlighted, and detailed point-by-point responses are provided below.

.

Comment:

The manuscript describes a compelling case report of a clinical intervention using a Noncontact Gesture-based Switch (AAGI) to enhance communication and social function in an individual with advanced Duchenne Muscular Dystrophy (DMD). The work is of interest to the rehabilitation medicine and assistive technology (AT) communities and describes an innovative and generally low-level posture of support for a population with extremely limited options for movement.

I believe the finding of a doubling of input over the 3-month period is a strong quantitative data collection. The suggested revisions below are intended to enhance the manuscript’s coherence and theoretical backing prior to publication. The authors have provided a unique, effective clinical intervention, and these revisions will help solidify the work’s place within the AT literature.

Specific Comments by Section

1Introduction

To enhance the theoretical context, the authors can briefly relate the need for the AAGI to the Technology Acceptance Model (TAM). For advanced DMD patients, the Perceived Ease of Use (a key component of TAM) for current switches (i.e., einp-and-puff or conventional button switches) becomes very low due to muscle weakness. This framing represents a robust academic foundation.

1-1.The existing literature review is fine but should also include recent works addressing non-contact/minimal effort interfaces in other similar rapidly progressive neuromuscular diseases (for example, advanced ALS) for the purpose of benchmarking the novelty of such an application in DMD.

Response:

We appreciate for your suggestion. We have reviewed recent papers concerning non-contact/minimal-effort interfaces for patients with ALS and related disorders and incorporated them into the manuscript. (L74-78 reference 2-4)

Specifically, in end-stage ALS, eye-tracking and visual BCI are being widely investigated in clinical practice (Edughele, 2022; Verbaarschot et al., 2021), and non-contact/low-burden alternative modalities such as single-switch optimisation have also been reported (Bonaker N 2023). By referencing these as benchmarks, this paper clarified the positioning of AAGI when applied to DMD

------------------------------------------

Comment:

2Case Presentation and Methods

The authors are to be commended for having selected an appropriate methodological framework (Case Report) to introduce this new intervention. Furthermore, the objective measurement of input speed is a methodological sound approach. To ensure maximum replicability (again for the AAGI and the input speed for a given condition) and scientific rigor, I suggest the following methodological refinements:

2-1Technical and Calibration Detail: The authors have fully described the AAGI system and the patient in detail. For replicability, can you provide more detail on the technical specifications (e.g., minimum camera resolution, type of vision algorithm)? Along those lines, please describe how you established the movement threshold to differentiate minimal voluntary from involuntary spasms or tremor, which are common in these conditions?

Response:

 We appreciate the reviewer’s insightful comment regarding the technical specifications and calibration procedures.

The AAGI system used in this study was developed based on the platform described in our prior works (see Method section of [Ref. 11,12]). Below, we provide detail for clarity and reproducibility.

Hardware Specifications:

The AAGI employs an Intel® RealSense™ Depth Camera D415, which integrates a stereo depth sensor with a minimum RGB resolution of 1920 × 1080 pixels and depth accuracy of ±2% at 2 m distance. The optical properties (field of view, depth precision, frame rate) and environmental constraints (illumination and background tolerance) are therefore dependent on the built-in specifications of this device.

Software and Gesture Detection Algorithm:

Gesture detection and classification were implemented using the custom software described in our previous methodological report (see Method section of [Ref. 11,12]). The system extracts spatiotemporal depth features from range images and classifies voluntary gestures using a trained adaptive thresholding algorithm.

Calibration and Movement Threshold Setting:

To differentiate minimal voluntary gestures from involuntary spasms or tremor, we employed an individualized calibration procedure. Each user performed 10 repetitions of intended movement, from which the mean amplitude and velocity of voluntary motion were computed. The detection threshold was then set slightly above the 95 percentile of background.This approach has been validated in prior internal testing and was refined during clinical deployment to optimize stability and reduce false-positive activations.

These details have been added to Section 2.3 (line 157-183)

-----------------------------------------

comment:

2-2Training Protocol: Please provide clarification on the training protocol – how often were they trained, what duration did training typically entail, and were any specific reinforcement methods used to enforce consistency in the activation gesture?

Response:

We appreciate the reviewer’s comment. The AAGI system requires minimal user training because it operates as an intuitive noncontact switch once the initial calibration is completed. The setup process involves identifying an optimal camera position and a reliable gesture source (e.g., cheek movement), which are determined collaboratively by the user, a caregiver, and a clinician familiar with the patient’s motor abilities. After this initial calibration, no structured training or reinforcement was necessary. The participant was able to use the system immediately as a standard binary switch. Minor adjustments to camera position or sensitivity were made only when the participant’s motor pattern changed or at their request to improve comfort and responsiveness.(lines 190-198)

-----------------------------------------

comment:

2-3Baseline Measures: Please clarify the method used for baseline assessment. What specific standardized communication or Quality of Life (QoL) measures were used to establish the pre-AAGI status? If specific measures were utilized, naming them, if appropriate, would add rigor to the pre-post assessments and subscription outcomes.

Response:

We appreciate for this helpful comment. We have clarified the assessment schedule and measures used. Prior to AAGI introduction (pre-AAGI), baseline evaluations included physical function and quality of life (QoL) using the SF-36. After initial device use (post-AAGI), the System Usability Scale (SUS) was administered to evaluate usability. At the three-month follow-up, both the SF-36 and SUS were re-administered to assess longitudinal changes in function, QoL, and user experience. These details have been added to Section 2.4 (“Outcome Measures” line 211-214) to improve clarity.

-----------------------------------------

comment:

3Results

3-1The presentation of the speed metrics for input is excellent. You might approach the QoL and social function results more forcefully by making better use of qualitative data. Including patient or caregiver quotes that were de-identified and specific helped with the anecdotal outcome of social activity change (e.g., "was able to complete a purchase online independently," "had the opportunity to participate in a group chat for the first time").

Response:

We appreciate this constructive suggestion. In response, we have added qualitative observations from both the participant and caregiver to illustrate changes in daily activities, social engagement, and motivation following AAGI implementation. These anonymized narratives were derived from post-intervention interviews and are now incorporated in the Results section. They complement the quantitative measures (SUS and SF-36) by providing a richer understanding of the user’s functional and psychosocial improvements(3.4 Narrative Impressions line 342-354).

-----------------------------------------

comment:

3-2Also, ensure that when providing figures around speed change, it is easy to identify the baseline and final measurement. A simple line graph that depicts the trajectory of improvement over the three-month period or reminds readers that a learning curve has been created would show learning from pre to post.

Response:

We appreciate for your suggestion. We agree that a time-based trajectory of input performance would enhance interpretability. However, the immediate post-calibration data were unavailable due to a technical recording issue during that session. Accordingly, we present the available pre- (baseline) and post- (3-month) data points, which still demonstrate a clear and meaningful improvement in characters-per-minute (CPM) performance. This clarification has been added in the revised Resultssection (Section 3.3 lines 323-326).

-----------------------------------------

comment:

4Discussion

This is the area where the most robust academic significance can be generated. The authors should definitely consider linking the outcome in practice to the applicable theory.

4-1Connection between Theory and Practice - Social Model: A substantial addition would be to relate the success of the AAGI to the Social Model of Disability. The technology served as a crucial environmental modification, removing a physical barrier (the requirement for switch contact) to digital and social participation. The intervention shifts the focus of the impairment (muscle weakness) away from the individual to an environmental barrier, thus enhancing Participation, a core outcome in the ICF classification system.

Response:

We appreciate this valuable suggestion. We have revised the Discussion (Section “Clinical significance: quality of life and participation”) to explicitly relate the findings to the Social Model of Disability and ICF principles. The new text highlights that the AAGI acts as an environmental modification, removing physical barriers and enabling participation rather than compensating for impairment. This emphasizes the alignment between technological and social perspectives on disability.(lines 449–459)

-----------------------------------------

comment:

 4-2Compare Emerging Trends - Niche: The discussion can further compare the AAGI's role to the considerably more complicated higher cost emerging trends/systems like Brain-Computer Interfaces (BUI) or highly advanced Eye-Tracking Systems. The AAGI is an extremely important non-invasive option in the ‘low-effort, low-cost, high-fidelity’ niche. It is a critical starting point for patients that have so little remaining movement to physically operate a switch, yet still express eye fatigue (thus are unable to perform eye-tracking movements) or do not have BUI options nor the ability to interface with one.

Response:

We have expanded the section “Comparison with other AAC modalities” to compare AAGI with eye-tracking systems and BCIs, clarifying its role within the “low-effort, low-cost, high-fidelity” niche. The revised discussion now emphasizes the advantages of simplicity, accessibility, and minimal fatigue relative to these complex technologies, while situating AAGI as a pragmatic and inclusive solution for patients with advanced neuromuscular conditions.
(lines 429–446)

-----------------------------------------

comment:

4-3Implications Telerehabilitation - In Implications it should discuss the anticipation that this camera interface could be easy to integrate into Tele-Rehabilitation models. The set-up includes visual tracking and requires minimal physical interaction with the patient, which would allow for the remote calibration, tuning, and potentially monitoring of switch set-up parameters, a promising future application for the home-care segment.

Response:

We agree that this is an important and timely point. Accordingly, we have added a dedicated paragraph in the “Limitations and Future Perspectives” section discussing telerehabilitation integration. The new text highlights that the non-contact camera-based interface allows remote calibration, tuning, and monitoring, enabling clinicians to support users from a distance. We also note that AAGI could serve as a component of home-based digital rehabilitation and telehealth ecosystems, facilitating longitudinal assessment and reducing service disparities. (lines 498–510)

-----------------------------------------

comment:

5 Conclusions

The Conclusion should be re-evaluated for emphasis and provide a clear, prescriptive recommendation. Wording should be suggested to establish AAGI as an important consideration in a continuum of care for patients with progressive neuromuscular diseases when the patient reaches the point of progressive functional decline where other input options can no longer be used.

Ans:

We appreciate this insightful suggestion. We have substantially revised the Conclusion to clarify the clinical and practical positioning of the AAGI within the continuum of care for individuals with progressive neuromuscular diseases. The revised paragraph now emphasizes that noncontact gesture-based switching can meaningfully extend independence and social participation, even at advanced stages of motor decline.

At the same time, we sought to frame AAGI not merely as a “last-resort” option but also as an augmentative and complementary tool that may be introduced earlier in the disease course to support ongoing engagement. To maintain a balanced and realistic tone, we refined the final statement to acknowledge that while advances in medicine and technology increasingly allow preservation of residual motor function, AAGI can serve as a practical complement—rather than a speculative replacement—within assistive technology ecosystems.

This adjustment underscores both the current clinical relevance and the forward-looking potential of noncontact interfaces, aligning with your intent for a conclusion that is prescriptive, grounded, and meaningful in the continuum of care (lines 516-525).

Reviewer 2 Report

Comments and Suggestions for Authors

The quality of the manuscript is relatively good and the description is also relatively clear. The reviewer has the following issues for the authors to consider.

  1. In Section 2.2, the study adopts aninput test duration of 2 minutes. If possible, please give more explanations on the rationale for this choice and its relevance to practical communication scenarios. Also, it would be helpful to clarify whether the ‘characters per minute (CPM)’ metric is derived from a single test or averaged across multiple measurements.
  2. In Section 2.3, the description of the AAGI device is overly brief. The authors should provide additional technical details, including the sensor model, sampling frequency, signal processing algorithm, and sensitivity calibration method, to enhance the reproducibility of the study. The criteria for recognizing effective gestures should also be specified, including motion amplitude, detection distance, response latency, and false activation rate.
  3. In Section 2.4.2, the manuscript states that the original item responses and excluded questions are presented in ‘Supplementary Table S1’,whereas the appendix lists this content under the title ‘Scheme 2’. The authors should unify the naming or clearly indicate in the main text that this material appears in the appendix to avoid confusion and ensure consistency in manuscript structure.
  4. In Figure 1 of Section 3.1, the pre- and post-intervention raw scores and calculated scores for items Q4 and Q10 change in opposite directions. Since the reverse scoring rule is not described, the authors should specify which items are reverse-scored and explain the calculation procedure in the Methods section to avoidreader confusing or misunderstanding.
  5. Throughout the manuscript, all resultsincluding tabular data such as Figures 1 and 2 are labeled as ‘Figures’. The authors should confirm with the editorial office during final preparation whether figures and tables should be distinguished in labeling to maintain consistency and clarity in the final layout.
  6. The overall reference formatting is appropriate; however, some entries lack complete publication information, including the year, volume, page numbers, or publisher details (for example, references [3], [26], and [37]). The authors should verify all references, add the missing publication information, and ensure accuracy and consistency in the reference list.

Author Response

We greatly appreciate your constructive suggestions, which have helped us to substantially improve the clarity, methodological rigor, and overall quality of the paper.

All comments were carefully considered and addressed in the revised manuscript. Revisions are highlighted, and detailed point-by-point responses are provided below.

Comment:

The quality of the manuscript is relatively good and the description is also relatively clear. The reviewer has the following issues for the authors to consider.

1.In Section 2.2, the study adopts aninput test duration of 2 minutes. If possible, please give more explanations on the rationale for this choice and its relevance to practical communication scenarios. Also, it would be helpful to clarify whether the ‘characters per minute (CPM)’ metric is derived from a single test or averaged across multiple measurements.

Response:

We appreciate the reviewer’s careful reading and constructive suggestion. The 2-minute test duration was selected considering the participant’s high fatigue level due to advanced-stage neuromuscular weakness. While 10-minute trials are often used as a standard in text-entry evaluations, a shorter 2-minute duration has been adopted in prior AAC studies as a practical compromise for individuals with severe motor impairment, ensuring data reliability while minimizing participant burden [reference 9,10].  The CPM value was calculated from a single 2-minute test to minimize participant burden and prevent fatigue-related performance variability. The revised text in lines 125-132)

-----------------------------

Comment:

2.In Section 2.3, the description of the AAGI device is overly brief. The authors should provide additional technical details, including the sensor model, sampling frequency, signal processing algorithm, and sensitivity calibration method, to enhance the reproducibility of the study. The criteria for recognizing effective gestures should also be specified, including motion amplitude, detection distance, response latency, and false activation rate.

Response:

We appreciate the reviewer’s valuable suggestion. In the revised Section 2.3 (Methods), we have substantially expanded the technical description of the AAGI device. Specifically, we have added the sensor model (Intel RealSense D415), sampling frequency (60 fps), and details on the signal processing algorithm used for gesture recognition. Furthermore, we have elaborated on the sensitivity calibration method, including quantitative criteria for gesture recognition (motion amplitude, detection distance, response latency, and false activation rate). These details have been added on lines 157-198 in the revised manuscript to enhance reproducibility and clarity.

-----------------------------

Comment:

3.In Section 2.4.2, the manuscript states that the original item responses and excluded questions are presented in ‘Supplementary Table S1’,whereas the appendix lists this content under the title ‘Scheme 2’. The authors should unify the naming or clearly indicate in the main text that this material appears in the appendix to avoid confusion and ensure consistency in manuscript structure.

Response:

We appreciate for pointing out this inconsistency. We have corrected the labeling of the supplementary material to ensure consistency throughout the manuscript.
Specifically, the section previously titled “Scheme 2” has been renamed “Supplementary Table S1,” consistent with the reference in Section 2.4.2.
The revised text now clearly indicates that the material appears in the appendix (line 565 supplementary Table S1

-----------------------------

Comment:

4.In Figure 1 of Section 3.1, the pre- and post-intervention raw scores and calculated scores for items Q4 and Q10 change in opposite directions. Since the reverse scoring rule is not described, the authors should specify which items are reverse-scored and explain the calculation procedure in the Methods section to avoid reader confusing or misunderstanding.

Response:

We appreciate your careful reading and helpful comment. In the revised manuscript, we have clarified the scoring procedure of the System Usability Scale (SUS) in the Methods section (Section 2.4.1). Specifically, we now indicate that odd-numbered items (Q1, Q3, Q5, Q7, Q9) were scored as the response value minus 1, while even-numbered items (Q2, Q4, Q6, Q8, Q10) were reverse-scored (5 minus the response value), following the standard SUS scoring method (Brooke, 1996). We also note that the SUS items are intentionally arranged so that odd-numbered statements reflect positive user perceptions (e.g., ease of use, confidence), whereas even-numbered statements reflect negative perceptions (e.g., complexity, inconsistency). This alternating structure ensures response balance and minimizes acquiescence bias.
This addition clarifies why items such as Q4 and Q10 show changes in opposite directions and prevents potential reader confusion (lines 220-223).

-----------------------------

Comment:

5.Throughout the manuscript, all results including tabular data such as Figures 1 and 2 are labeled as ‘Figures’. The authors should confirm with the editorial office during final preparation whether figures and tables should be distinguished in labeling to maintain consistency and clarity in the final layout.

Response:

We thank the reviewer for this helpful observation. In the revised manuscript, all items that present data in tabular format have been relabeled as “Tables” instead of “Figures” to ensure clarity and consistency throughout the manuscript. This change has been applied to both the text and the corresponding captions (e.g., Figures 1 and 2 are now labeled as Tables 1 and 2).
We appreciate the reviewer’s suggestion, which has improved the manuscript’s overall readability and structure(Lines 279,312)

-----------------------------

Comment:

6.The overall reference formatting is appropriate; however, some entries lack complete publication information, including the year, volume, page numbers, or publisher details (for example, references [3], [26], and [37]). The authors should verify all references, add the missing publication information, and ensure accuracy and consistency in the reference list.

Response: We appreciate your suggestion. We vertified the missing informations as follows: (reference number has been changed due to adding new references due to the suggestion of reviewer 1)

3->5 Li Yu HA, Md Zobaer Islam, John F. O’Hara, Christopher Crick, Sabit Ekin. Hand Gesture Recognition through

Reflected Infrared Light Wave Signals. 10th International Conference on Electrical and Electronics Engineering (ICEEE) 2023 doi: 10.48550/arXiv.2301.05955

26->29 Majaranta P, Räihä K-J. Twenty years of eye typing: systems and design issues. Proceedings of the 2002 symposium on Eye tracking research & applications. New Orleans, Louisiana: Association for Computing Machinery, 2002:15–22.

37->40 Albert M. Cook P, PE (ret), Janice Miller Polgar, BScOT, PhD, FCAOT and Pedro Encarnação, PhD, Ha HS. Chapter8 Control Interfaces for Assistive Technologies  Assistive Technologies, 5th Edition Elsevier 2020.